# Distinctive Features of Graphene Synthesized in a Plasma Jet Created by a DC Plasma Torch

**DOI:** 10.3390/ma13071728

**Published:** 2020-04-07

**Authors:** Marina Shavelkina, Peter Ivanov, Aleksey Bocharov, Ravil Amirov

**Affiliations:** Joint Institute for High Temperatures of Russian Academy of Sciences, 125412 Moscow, Russia; mshavelkina@gmail.com (M.S.); peter-p-ivanov@yandex.ru (P.I.); amirovravil@yandex.ru (R.A.)

**Keywords:** synthesis, graphеne flakes, plasma jet, conversion of hydrocarbons, hydrogen, quasi-one-dimensional flow

## Abstract

Synthesis of graphene materials in a plasma stream from an up to 40 kW direct current (DC) plasma torch is investigated. These materials are created by means of the conversion of hydrocarbons under the pressure 350–710 Torr without using catalysts, without additional processes of inter-substrate transfer and the elimination of impurities. Helium and argon are used as plasma-forming gas, propane, butane, methane, and acetylene are used as carbon precursors. Electron microscopy and Raman imaging show that synthesis products represent an assembly of flakes varying in the thickness and the level of deformity. An occurrence of hydrogen in the graphene flakes is discovered by X-ray photoelectron spectroscopy, thermal analysis, and express-gravimetry. Its quantity depends on the type of carrier gas. Quasi-one-dimensional approach under the local thermodynamic equilibrium was used to investigate the evolution of the composition of helium and argon plasma jets with hydrocarbon addition. Hydrogen atoms appear in the hydrogen-rich argon jet under higher temperature. This shows that solid particles live longer in the hydrogen-rich environment compared with the helium case providing some enlargement of graphene with less hydrogen in its structure. In conclusion, graphene in flakes appears because of the volumetric synthesis in the hydrogen environment. The most promising directions of the practical use of graphеne flakes are apparently related to structural ceramics.

## 1. Introduction

According to the Union of Pure and Applied Chemistry (UPAC) definition graphene layer is a single carbon layer of the graphite structure; its nature being an analogy to a polycyclic aromatic hydrocarbon of quasi-infinite size [1]. In contrast to ideal graphene with regular arrangement of carbon atoms the experimental graphene has deformities resulting in several physical forms: flakes, bands, leaves, and some others [2]. In the scope of the mentioned forms, the graphene might have one, two, or several (≤10) sheets of carbon atoms with sp2-hybridization. Different distributions of the benzene rings in the graphene sheet determine the shape, size, edges, and number of layers and additional covalent or noncovalent bonds with other atoms which result in modifications of the electrical or chemical properties of graphene [3], as well as various application modes. That is why graphene properties depend on the initial chemical composition of precursor and on the environment during the formation. It is demonstrated in [4], that the graphene becomes doped with nitrogen atoms, if there is nitrogen in the precursor or in the carrier gas. When solid precursors like polymers, food rests, insects, wastes, and so on are used it is possible to create monolayer or double-layer graphene in the H_2_/Ar stream directly on the metal catalyst. Its properties are similar to those of the graphene created by means of chemical vapor deposition (CVD); existing peculiarities are due to type of precursor [5]. Graphene synthesized on wide area and transferred to the metallic nets to be used as transparent-sensing screens in personal electronics and in photoelectric devices has properties dissimilar to those when it is on the copper foil [6]. In order to avoid the property change of graphene because of its transfer from one surface to another, to make it less expensive, and to increase the output in the range of required characteristics one has to grow the graphene directly on the substrate under consideration [7], to make the figured graphene applicable in transistors [8] or to make graphane/graphene structure in the scope of one sheet [9]. Nowadays various methods for the treatment of transferred sheets with hydrogen are under development. Researchers from Nanjing University in China [10] have used a plasma-enhanced CVD method to make large graphene films free of any wrinkles. The new method allows developing ultra-flat graphene sheets on a copper substrate of about 10 cm in size in several minutes using hydrogen plasma. The ultra-smooth films could enable large-scale production of electronic devices that harness the unique physical and chemical properties of graphene and other 2D materials. The controllability of the synthesis process in order to get graphene with desired properties continues to be an unsolved problem as of today. 

Graphene synthesis in the plasma jets is a fast process without catalysts, there is no need of intermediate processes, in outer heating of substrate. Regardless of the type of plasma generator there are several advantages. First, high temperature in the plasma reactor provides nearly full dissociation of injected reagents which helps to form high temperature materials in the vaporized state. Second, high energy density of thermal plasma provides a high productivity even in small reactors. Finally, in the plasma process a fast quenching is possible, allowing the high degree of overcooling and supersaturation, which is favorable for the formation of nanoparticles [11,12,13].

Shortcoming of plasma processes is existence of cool boundary layers and non-uniformity of process parameters. Girshick et al. [14] have developed numerical model of nucleation and growth of particles in reactor. They have found that the particle formation is greatly dependent on two-dimensional non-uniformity of temperature and velocity profiles [14]. According to Y. Tanaka inhomogeneity conditions in plasma accelerate the synthesis of nanosized particles [15]. Thus, there is a possibility to control the nucleation of nanoparticles and their properties by means of profiles of temperature and velocity [16].

Graphene synthesized by means of microwave plasma, high-frequency induction plasma, or DC plasma torch, as a rule, has a unique structure—flakes with number of sheets from 1 up to 10 and more [17,18,19]. On the contrary, graphene synthesized on the substrate makes rather smooth film with low number of sheets [20,21,22]. Precursors in these cases may be the same—methane, ethanol, for example. Analysis of the processes in the carbon nucleation zone of the plasma jet generated by the plasma torch indicates high concentration of the atomic hydrogen, hydrocarbon radicals, or the C6H6-like compounds; these substances, according to [23,24], might promote the graphene formation. 

This paper discusses the influence of the DC plasma composition on the graphene morphology. According to references [25,26] the thermal decomposition of different hydrocarbons in the high temperature range is limited by the reaction of dissociation of methane to carbon and hydrogen, and an appreciable hydrogen output occurs at 1620–1670 К. In our experiments, the temperature varied from 8000 K (at the anode nozzle) to 1200 K (at the outlet of cylindrical reactor), further on the reactor space expands, and gas temperature falls dramatically down to 700–800 К. Fast synthesis and low temperatures tend to create in the plasma stream flake-like structures and hydrogen. According to Tesner P.A. [27] the carbon condensed in the free volume must have a spherical shape. It does not agree with our experiments. Apparently, the formation of nanoparticles is not limited by dehydrogenation reactions. There is an opinion [28,29] that the condensed carbon appears during the thermal decomposition of hydrocarbons in the free volume because of the chemical reactions as a result of disintegration of intermediate compounds. It is confirmed by the fact that in the heated hydrocarbons (above the nucleation of carbon particles) there are some unsaturated hydrocarbons, acetylene, among them that are easily released from hydrogen. Therefore, there must be an interconnection between the intermediate products in plasma and the graphene configuration.

## 2. Methods 

Layout of the setup for the graphene synthesis is shown in Figure 1. It is a conventional plasma chemical reactor with water-cooled casing where the end product is manufactured by means of plasma jet consisting of neutral gas and the precursor of desired substance. The peculiarity is the synchronous input of plasma forming gas and carbon precursor into discharge gap of the plasma torch and the arrangement of the collector of the end product. DC plasma torch with power input up to 40 kW was used as a generator of plasma jet [28]. Its original features are an expanding channel of the outlet electrode and the swirling-type stabilization of plasma jet. The design provides a uniform velocity distribution, homogeneity of parameters, and the arc stabilization along the whole length of the positive column, a stable functioning of installation and a better heat transfer at the beginning of the arc. The installation includes, besides of plasma torch, its power supply, the water cooling system, the graphite insertion, defending the metallic surface of the transitional area against plasma jet, vacuum chamber, degassing and injection systems, rotameters, containers of plasma forming gases and carbon precursors. Easily available commercial hydrocarbons with varying relation of carbon to hydrogen were used as precursors: propane, butane, acetylene, and methane. Helium, marked A, and cleaned argon were used as the carrier gases. The essence of the synthesis method is the decomposition of hydrocarbons in the argon or helium plasma jet and subsequent quenching of the resulting vapor in the reactor and formation of the solid deposition on the inner side walls of the reactor and on the surface perpendicular to the plasma jet. Variation of pressure in the range 80–740 Torr by means of simplest liquid-packed ring pump controls the arc voltage under the constant current value. The current value is chosen on the sharp increase part of the current-voltage characteristics of the plasma torch.

Table 1 presents the parameters of experiments regarding the multilayered graphene flakes synthesis. The synthesis process duration equals to 6 min and is governed by the quantity of specimens required to investigate their properties. Total output of the carbon sediment collected on the target varies from 6 g to 10 g (99 wt. %).

We measured the temperature in the zone of carbon vapor condensation and formation of graphene and graphane by the chromel-alumel thermocouple. Depending on the experimental conditions, the temperature on the collector surface varied within the range of 700–1200 К. This temperature range is optimal to obtain the hydrogenated graphene [30,31].

Samples are obtained as black powder with apparent density up to 10^−8^ g/cm^3^ and are investigated at the initial state after gathering in the cooled reactor. For the immediate registration of dimensional parameters and image data a scanning electron microscopy (SEM) with a Hitachi S5500 microscope (Hitachi High-Technology Co., Tokyo, Japan) in the DF-STEM, BF-STEM, and SE modes was used. Samples were identified by means of a Raman spectroscopy study (Ntegra Spectra, NT-MDT Spectrum Instruments, Zelenograd, Russia), a laser with a wavelength of 532 nm). In order to quantitatively investigate the elemental composition and the electron state of the atoms, we used the X-ray photoelectron spectroscopy (XPS), the measurements were performed at room temperature by means of the PHOIBOS 150 hemispherical energy analyzer (SPECS GmbH, Berlin, Germany); we took the Mg Kα (12.5 kV, 250 W) emission without using the system of the surface charging neutralization. The elemental composition was determined by express-gravimetry (vario MICRO cube). The method is based on the pyrolytic burning of the substance in an empty tube in the scope of the quartz container with hanging washed by oxygen flow. After the burning, the compound with the element under consideration appears whose mass is to be measured. The phase composition of the synthesized samples was investigated by thermal methods (thermogravimetry and differentially-scanning calorimetry), under the linear heating with the rate of 10 K/min, in the argon ambient, by means of the thermo-analyzer STA 409PC Luxx with quadrupole mass spectrometer QMS 403 C Aeolos (Netzsch, Germany).

The X-ray structure analysis was performed using the standard technique and the DRON-2(Russia) facility (CuKα-emission). We obtained the X-ray diffraction spectra from the basic surfaces of the samples.

## 3. Results

Series of experiments were performed with the decomposition of propane and butane, methane, and acetylene in the helium plasma under the pressure 350, 500, and 710 Torr. Helium flow rate was kept constant at the level 0.75 g/s. The precursor flow rates are shown in Table 1. The greatest output of graphene occurs when the precursor is a mixture of propane and butane (70:30 mass %) in the range of flow rate 0.11–0.3 g/s – it runs up to 5 g/min. Images of materials synthesized in the propane-butane/helium plasma jet are presented in Figure 2. They show the structure morphology in the shape of flakes with lateral dimensions 200–500 nm which does not depend on pressure. Subjects to change are only lateral dimensions of flakes and their dispersion. Greatest degrees of equality of dimensions have the flakes synthesized under the pressure 500 Torr. The turned inwards edges of the structure are clearly visible in Figure 2a,c. It may be related to the boundary effect created by impurity atoms, which are ready to escape in the first place during the heating. It is in agreement with the investigations of thermal stability of samples. According to Figure 2d, which shows the mass change of sample because of the heating in the air, maximum output of CО_2_ occurs in the temperature range 820–920 K. It is worth to note that peak of the differentially scanning calorimetry (DSC) curve is produced not only by CО_2_, but also with output of other gases, among them may be pure hydrogen and hydrocarbon radicals like CH_3_, C_2_Hx, and C_3_Hx [31]. Results of X-ray diffraction study show that the material consists of stacks of equidistant graphite layers without crystallographic bonds. Interlayer gap d002 in these stacks is equal to 0.345 nm. Raman scattering spectra for the ensemble of flakes show three peaks—D, G, and 2D, which fall to 1334 cm^−1^, 1569 cm^−1^, and 2682 cm^−1^, respectively (Figure 3). Peaks D and G are high enough and not parallel to x-axis, as it is usually for the epitaxial graphene or the graphene produced by the scotch-method [32,33]. Besides, the peak D is quite intensive, and therefore for the sample presented in Figure 2c the relation I_D_/I_G_ is equal to 0.79, and for all samples synthesized from the propane-butane mixture this relation falls into the range 0.60–0.88. Common opinion is that in conventional systems with carbon in sp^2^-hybridization such as graphene, graphite, or carbon nanotubes, the lower the D/G, the lower the defect concentration (not taking into account doping or charging effects). But the hydrogenated graphene sp^2^-system is heavily disordered with considerable inclusion of sp^3^-system. Probably G-band is a superposition of G-band and D’-band. Besides the sample is an ensemble of randomly placed graphene flakes. As a result of all this there are more carbon atoms with sp^2^-hibridization around defects. 

By means of the direct express-gravimetry method, we determine the C:H content ratio in the samples. At the propane-butane mixture decomposition with the flow rate of 0.37 g/s in the helium plasma (710 Torr), we have obtained the hydrogen to carbon molar ratio of 1:4.

Minimal output of graphene material occurred in the case of methane with flow rate 0.2 g/min. Its injection into the helium plasma jet leads to the formation of flake-like structures with lateral dimension 200–1000 nm (Figure 4a–c). Figure 4d shows the increase in the amount of impurity gases in the DSC curve under low temperatures. 

X-ray analysis shows that spectra of the samples contain 2 weak halos. The first one is in the range of Bragg angles 2ϴ = 11÷12°. Maximum intensity of this halo corresponds to the interlayer gap d = 0.7694 nm. Second halo is in the range of angles 2ϴ = 24÷28°. Maximum intensity of this halo corresponds to the interlayer gap d = 0.3401 nm.

Figure 5 shows the Raman spectrum obtained in the study of ensemble of flakes. The spectrum shows three intense peaks corresponding to the D band at 1349 cm^−1^, the G band at 1586 cm^−1^, and the 2D band at 2693 cm^−1^. Ratio I_D_/I_G_ is equal to 0.86.

Analysis of the C1s carbon spectra shows that there are three ground states of carbon with the binding energies of 284.5 eV, 285.0 eV, and 291 eV. The spectra for the three states coincide within the domain of low binding energies and differ in the high-energy domain. These facts might be explained by surface shift of the C1s level and by the presence of defects in the surface sample site. The spectra differ in the width of the main peak because of the specimen non-uniformity caused by the defects and the different sizes of particles. The difference in the spectra within the lower binding energy region indicates different conductivity caused, possibly, by the presence of the C–H bonds. This fact is in agreement with [34].

The minimal quantity of the hydrogen atoms per a single carbon atom equals to 0.017.

Experiments with the graphene synthesis by means of addition of acetylene into helium plasma jet show that in the jet volume flakes are forming under the pressure 350–500 Torr (Figure 6a,b). Lateral dimensions of flakes are 200–700 nm. Under increased pressure of 710 Torr, graphene flakes are formed over dendritic structures (Figure 7a), and around some graphene particles one can observe a film, appearing because of polymerization of hydrocarbons under the influence of an electron beam (Figure 7a,b). In Figure 8 the result of thermal analysis of the ensemble of graphene flakes synthesized under 500 Torr is presented. The figure shows that samples lose the weight during disintegration in a rather narrow temperature range, and in the low temperature range (up to 673 K) loss of mass does not run over 5% (mass). Raman spectra of the sample obtained at 500 Torr (Figure 9) have most intensive D-peak, so the ratio I_D_/I_G_, which is equal to 0.96, clearly demonstrates the deformities of graphene flakes. 

Thus, experiments with the decomposition of propane, butane, methane, and acetylene show that typical morphology of graphene structures synthesized in the plasma jet is the shape of flake with bent edges. 

In order to assess the influence of carrier gas on the shape of graphene, experiments with argon were carried out. Repeated technological conditions were used in the case of helium, but argon flow rate was 3.5 g/s, and arc current was 350 A. These values are chosen on the increasing part of the current-voltage characteristics, the same as for helium. Figure 10a, Figure 11a, Figure 12a, show that the morphology of structures is constant in the same pressure range 350–710 Torr. Compared with samples obtained in the helium plasma argon allows the synthesis of greater graphene structures, especially using methane or soot. In the case of helium, big structures are synthesized when propane-butane mixture is used. According to thermal analysis (Figure 10b and Figure 11b) during the heating from 570 to 770 K active hydrogen output is visible. In Figure 13, Raman spectra of the samples obtained via the methane decomposition in argon plasma are presented. There are three characteristic peaks D, G, and 2D. The ratio I_D_/I_G_ is equal to 0.87. It is somewhat greater than in the case with helium.

C1s spectra have a conventional aspect with the main asymmetric peak at 284.4 eV and a satellite generated by excitation of π-π* transitions and π-plasmon (π-satellite), shifted from the main peak by 6.1 eV (Figure 14). Spectra differ in the width of main peak because of the different sizes of defects caused perhaps by the presence of hydrogen.

We investigated an influence of the addition of graphene flakes on the physical-mechanical properties of the structural ceramics based on the cubic boron nitride. Fabrication of industrial charge needed ~5 g of nanomaterials. This amount of pure graphene is obtainable only using the plasma chemical method of synthesis. The graphene with largest lateral size synthesized using argon/acetylene plasma was chosen. Ceramics were produced by means of the hot pressing (1800 К, 412 MТorr) and reinforced with carbon nanotubes, as one can see in Figure 15, at the fracture. Under the influence of temperature and pressure, the flakes are curled up to nanotubes, making the bending strength greater by 5%. It is known from the literature that the graphene, after the hydrogen is eliminated from one side, curls up to nanotube [35]. 

Thus, experiments with argon demonstrated as well that the formation of flake-like shape is related to the existence of hydrogen, but here particle dimensions are greater, and their dispersion is greater too.

One-dimensional approach to the plasma flow is an attempt to determine the processes initiating the formation of flakes during the synthesis of graphene in the volume of helium and argon plasma jets.

Common initial data for the comparison of helium and argon cases are the following: pressure 350 Torr, precursor – propane-butane mixture with flow rate 0.104 g/s.

Flow rates of helium and argon are 0.75 and 3.5 g/s correspondingly.

There are some motivations under the chosen figures. Helium flow rate is an optimum found experimentally; by the way it provides the carbon mass fraction in the plasma 0.1. Propane-butane mixture provides an intermediate value of H/C-ratio between methane and acetylene.

Differences in the cases of helium and argon are investigated using equilibrium plasma composition as a function of temperature in the range 2000–4000 K. As a matter of fact just in this range the precursor turns into molecular hydrogen and condensed phase of carbon, identified in the database IVTANTERMO [36] as graphite. Modeling of the plasma flow in the reactor is made using quasi-one-dimensional approach under the local thermodynamic equilibrium [28,37,38,39].

For the sake of simplicity the components with insignificant mass fractions are ignored in the composition of plasma, and ensemble of hydrocarbons (C_2_H, C_2_H_2_, C_2_H_3_, C_2_H_4_, C_4_H, C_4_H_2_, C_3_H) is presented with two items: bound carbon (CnH) and bound hydrogen (HCn). They are complement to the list of conventional components: C – carbon atoms, Cgr – condensed carbon, Cn – carbon molecules (C_2_, C_3_, C_4_, C_5_), H – hydrogen atoms, H_2_ – molecular hydrogen.

Mass fraction of ignored hydrocarbons in the considered temperature range is less than 1%.

In Figure 16 and Figure 17, results of the modeling of experiments with helium (0.75 g/s) and argon (3.5 g/s) are presented. The difference between two figures is evident. Maximum of the CnH curve in Figure 16 nearly coincides with the twist of Cn curve. The concentration of C is greater for helium. It is interesting that the concentration of H seems to be the same in two cases. In order to consider this issue in more details the hydrogen curves are plotted together in Figure 18. One can see that argon curves are shifted toward higher temperatures. Evidently in the argon environment the hydrogen concentration is less because of the more intensive escape of H_2_. The difference in the HCn content leads to the difference in the synthesized graphene particles. 

In Figure 19 and Figure 20, evolution of the molar composition of the argon plasma is presented in detail, in two ranges of concentration. Results of calculations are in agreement with the references [40,41,42]. Evidently the critical temperature of the beginning of condensation in this case is 3300 K. The formation of condensed carbon goes on at the expense of C, C_2_, C_3_, C_5_, C_2_H, C_2_H_2_, C_4_H_2_, and C_3_H. It can be assumed that hydrogen from C_3_H remains in the C-C structure and influences the shape of the future graphene structure.

In order to assess the effect of hydrogen on the shape of graphene, the experiments were selected producing graphene structures without impurities. Graphene quality was evaluated using the 5-point scale (Figure 21), with point 5 being awarded to graphene flakes with the most even surfaces. It happens that the shape nearest to the graphene synthesized by means of CVD method occurs when the partial pressure of hydrogen is smallest.

Thus, the presence of hydrogen in the graphene flakes demonstrates its key role in graphene synthesis, which is in agreement with a number of researches [43,44,45]. Probably H atoms can effectively cut off ragged carbon bonds creating carbon-hydrogen bonds, and to suppress the formation of spherical structure. Therefore, in the plasma jet, rich of hydrogen, flakes are formed. This formation allows the direct application of synthesized graphene, without additional treatment.

## 4. Conclusions

Volumetric synthesis of graphene materials under the pressure 350–710 Torr in a plasma stream from a high-power DC plasma torch is demonstrated. For various carbon precursors like propane, butane, methane, and acetylene and for helium and argon as carrier gas the graphene is synthesized usually in the form of flakes regardless of the varying input and pressure. It is found that during the heating of the graphene flakes synthesized in the plasma of any composition output gas always contains hydrogen. Results of Raman spectroscopy show a high content of defects in samples. Apparently, hydrogen is responsible for the bent form of graphene. Hydrogen content in precursor and the type of carrier-gas are shown to influence the lateral dimensions of graphene and its purity. Greatest output of pure graphene occurs when propane-butane mixture is added to the helium plasma jet, and maximal graphene dimensions are obtained using methane and argon. Thus, it is possible to control graphene properties by means of the plasma forming gas. Variation of temperature profile in reactor is in direct correspondence with the properties of plasma forming gas. At every instance, different plasma components are formed according to the temperature profile. Thermodynamic equilibrium calculations show that in the temperature range 2500–3500 K hydrocarbon radicals appear with varying C:H relation in correspondence with the falling plasma temperature, and there always is hydrogen. In the case of argon, the residence time of particles in the hydrogen-enriched atmosphere is greater, hence the lateral dimensions of flakes are also greater. In conclusion, it can be stated that hydrogen is responsible for the deformation of graphene layer to flake which is a distinctive feature of graphene synthesized in a plasma stream. The addition of synthesized flakes to nitride bore ceramics increased its mechanical properties. Therefore, the most promising directions of the practical use of graphene flakes are apparently related to structural ceramics.

## Figures and Tables

**Figure 1 materials-13-01728-f001:**
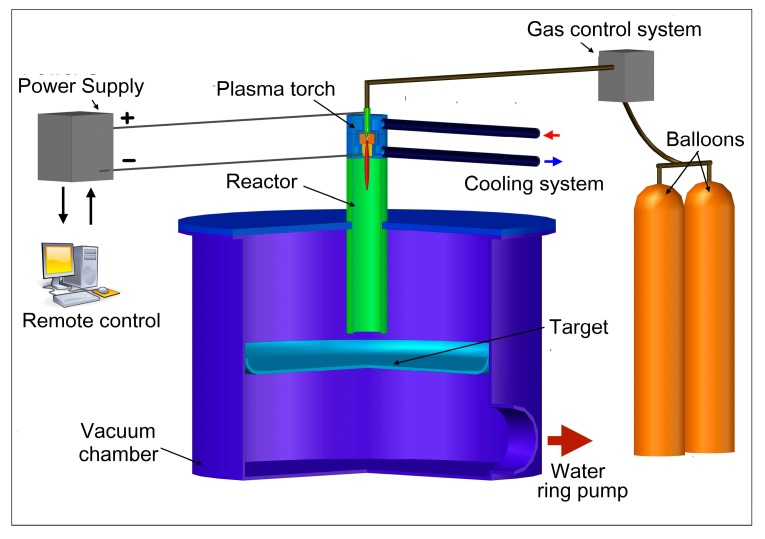
Schematic diagram of the experimental setup.

**Figure 2 materials-13-01728-f002:**
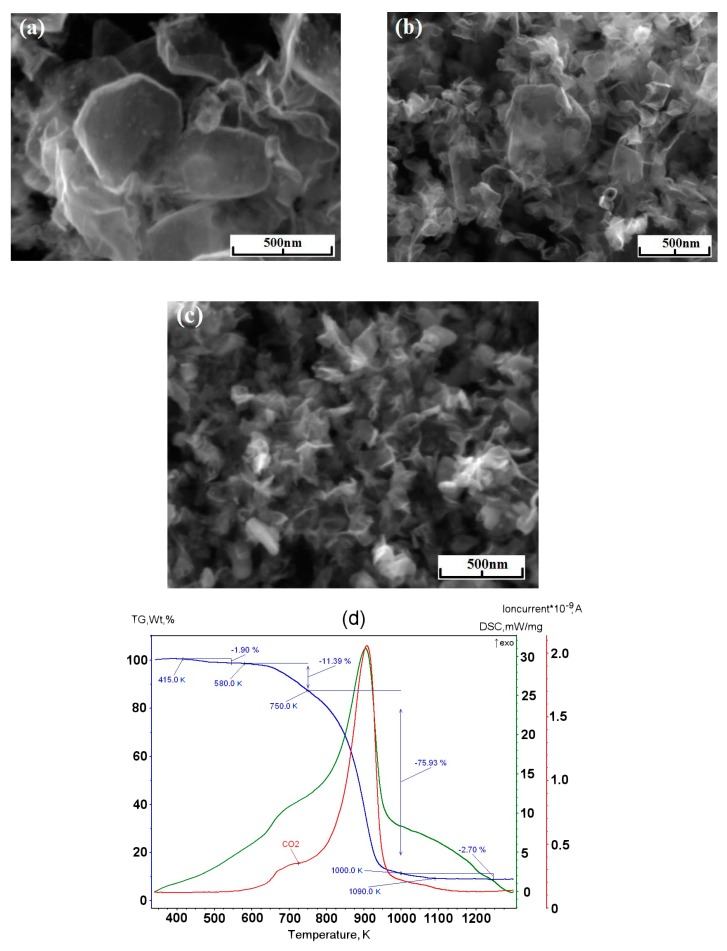
(**a**) BF-STEM image of the sample obtained via the propane–butane mixture decomposition in helium plasma at 350 Torr; (**b**) SEM image of the sample obtained via the propane–butane mixture decomposition in helium plasma at 710 Torr; (**c**) SEM image of the sample obtained via the decomposition in the helium plasma at 500 Torr; (**d**) typical result of thermal analysis of samples synthesized in propane–butane mixture /helium plasma at 350–710 Torr. Scale bars: (**a**–**c**) 500 nm.

**Figure 3 materials-13-01728-f003:**
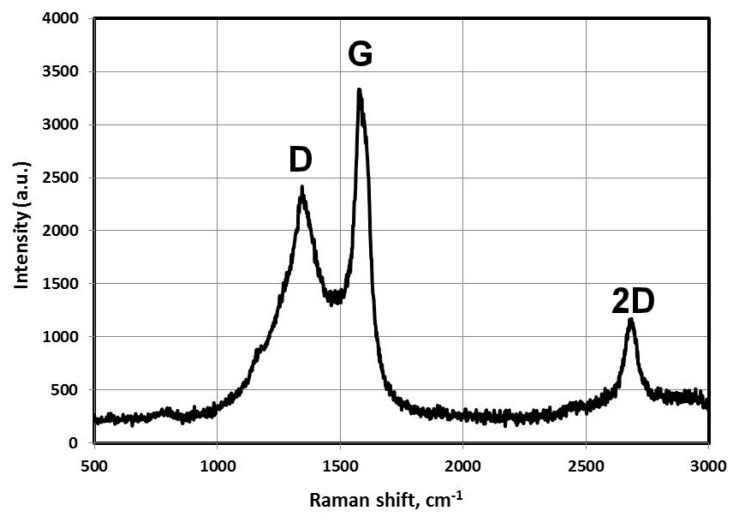
Raman spectra of the samples obtained via the propane–butane mixture decomposition in helium plasma at 350 Torr.

**Figure 4 materials-13-01728-f004:**
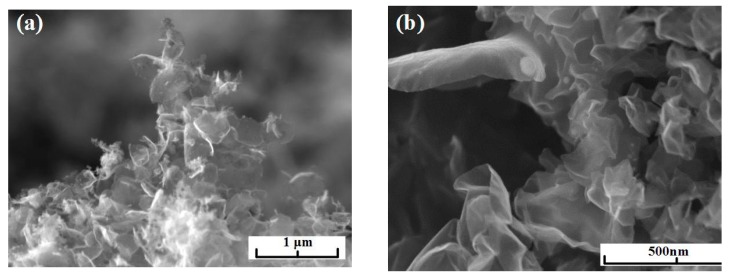
SEM image of the sample obtained via the methane decomposition in helium plasma: (**a**) at 350 Torr; (**b**) at 710 Torr; (**c**) at 500 Torr; (**d**) typical result of the thermal analysis of samples synthesized in methane/helium plasma at 350–710 Torr.

**Figure 5 materials-13-01728-f005:**
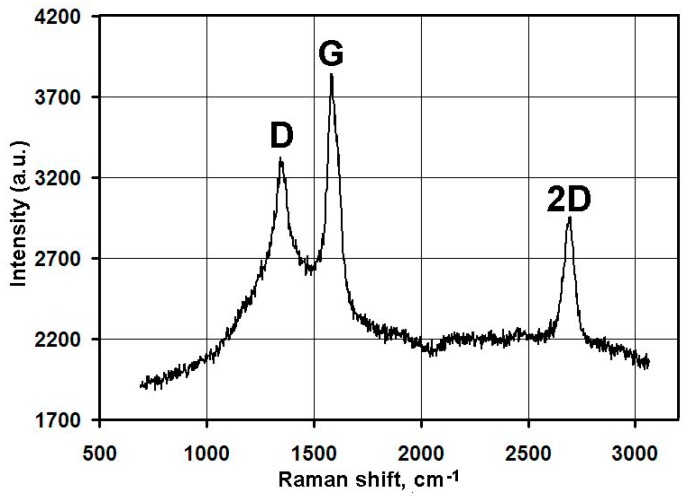
Raman spectra of the samples obtained via the methane decomposition in helium plasma jet at 500 Torr.

**Figure 6 materials-13-01728-f006:**
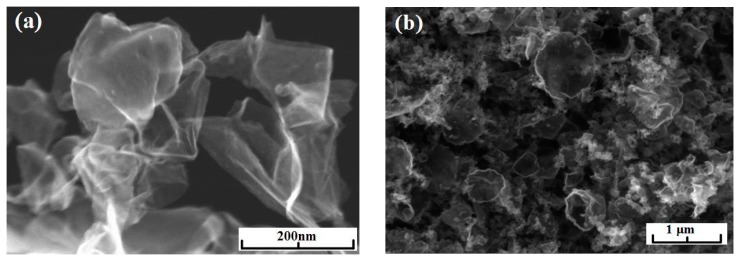
SEM image of the sample obtained via the acetylene decomposition in helium plasma: (**a**) at 350 Torr; (**b**) at 500 Torr.

**Figure 7 materials-13-01728-f007:**
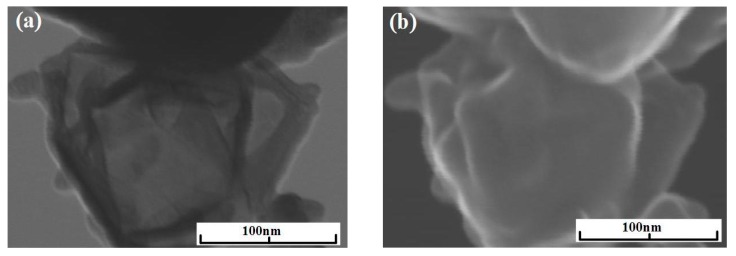
(**a**) BF-STEM image of the sample obtained via the acetylene decomposition in helium plasma at 710 Torr; (**b**) SEM image of the sample obtained via the acetylene decomposition in helium plasma at 710 Torr.

**Figure 8 materials-13-01728-f008:**
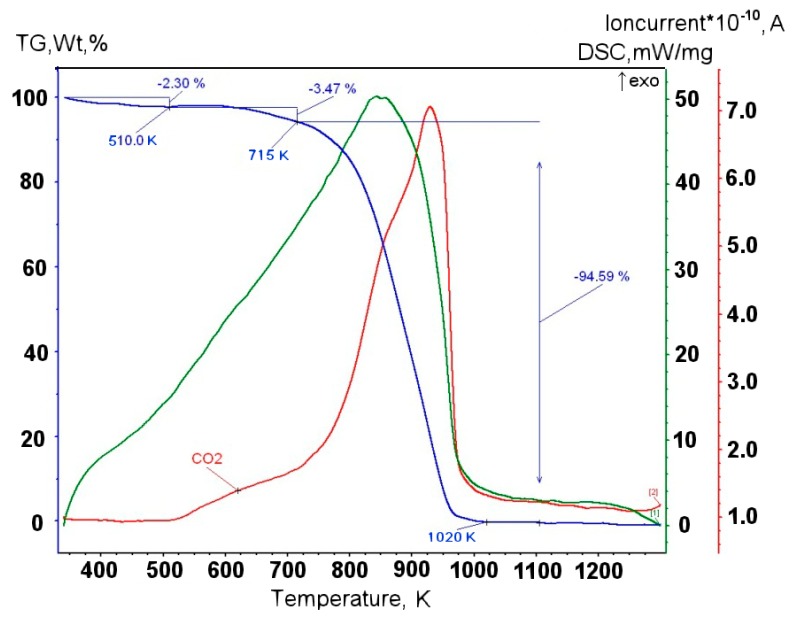
Thermal analysis of ensemble of graphene flakes, synthesized in acetylene/helium plasma at 500 Torr.

**Figure 9 materials-13-01728-f009:**
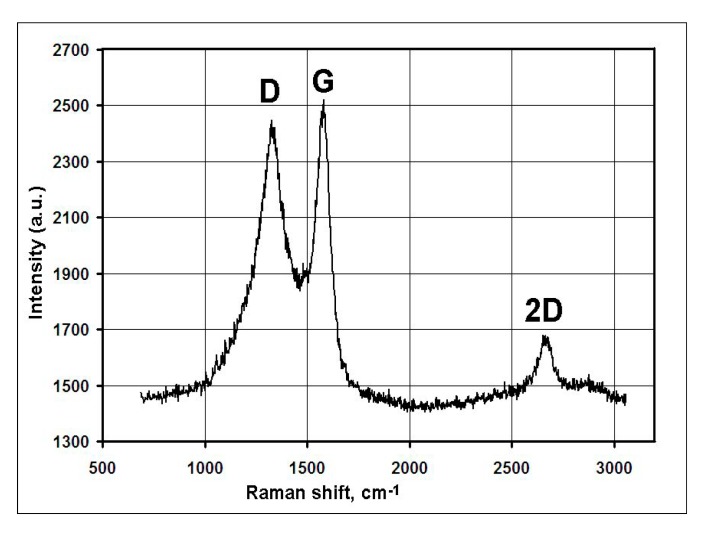
Raman spectra of the samples obtained via the acetylene decomposition in helium plasma jet at 500 Torr.

**Figure 10 materials-13-01728-f010:**
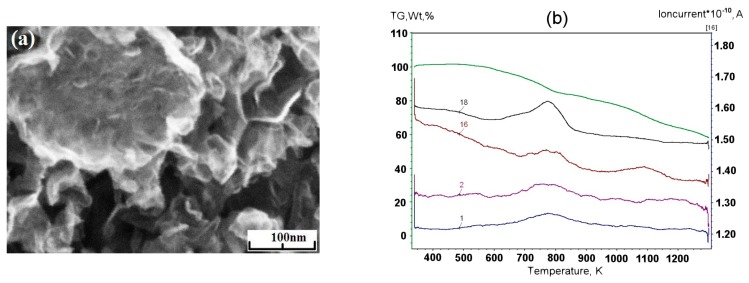
(**a**) SEM image of the sample obtained via the propane–butane mixture decomposition in argon plasma at 500 Torr; (**b**) the mass-spectrometry measurement of a sample obtained via the propane–butane mixture decomposition in argon plasma at 500 Torr. The numbers indicate molecular weights of the released gases: 1 amu -H, 2 amu -H_2_, 16 amu CH_4_, 18 amu -H_2_О.

**Figure 11 materials-13-01728-f011:**
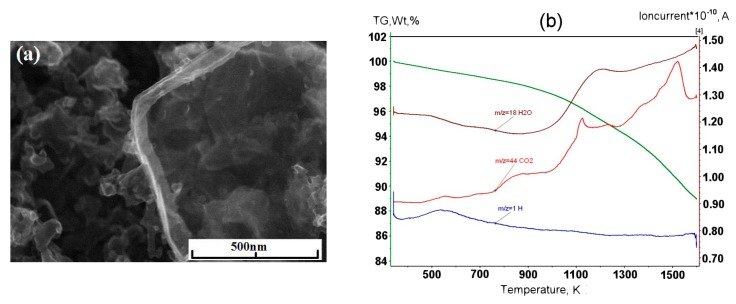
(**a**) SEM image of the sample obtained via the methane decomposition in argon plasma at 350 Torr; (**b**) the mass-spectrometry measurement of the sample obtained via the acetylene decomposition in helium plasma at 350 Torr.

**Figure 12 materials-13-01728-f012:**
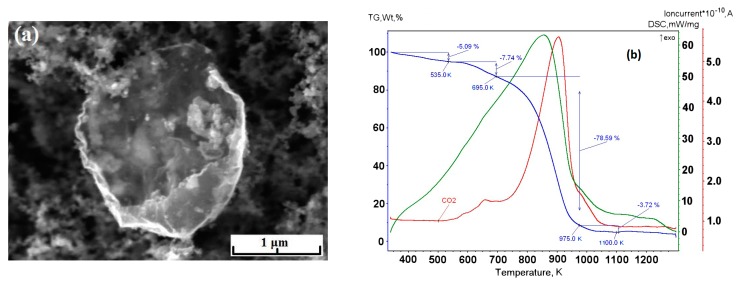
(**a**) SEM image of the sample obtained via the acetylene decomposition in argon plasma at 350 Torr; (**b**) thermal analysis of ensemble of graphene flakes, synthesized in acetylene/argon plasma at 350 Torr.

**Figure 13 materials-13-01728-f013:**
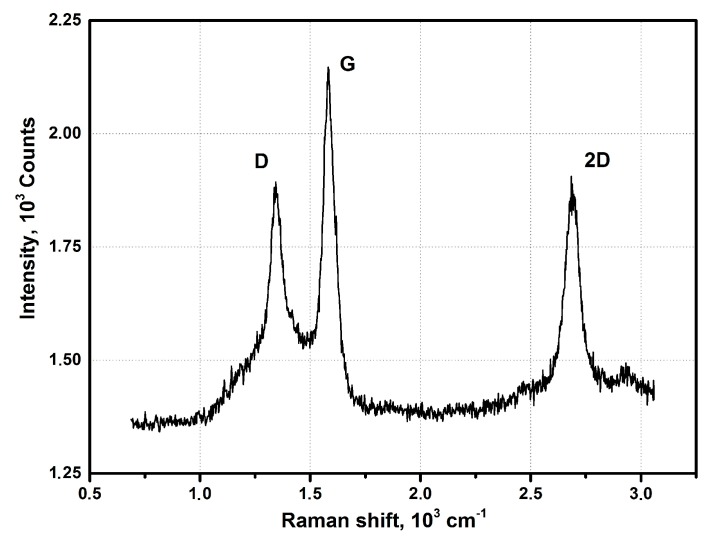
Raman spectra of the samples obtained via the methane decomposition in argon plasma at 350 Torr.

**Figure 14 materials-13-01728-f014:**
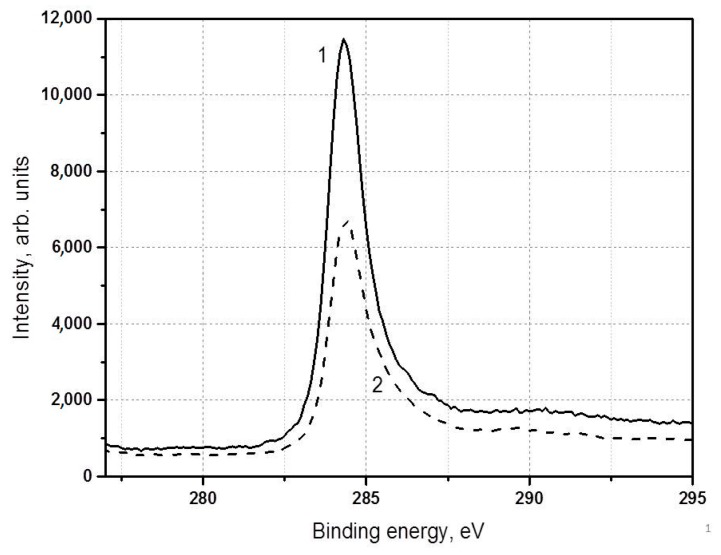
С1s spectra of the samples obtained in argon plasma via the methane decomposition at 500 Torr (1) and via the propane-butane decomposition at 710 Torr (2).

**Figure 15 materials-13-01728-f015:**
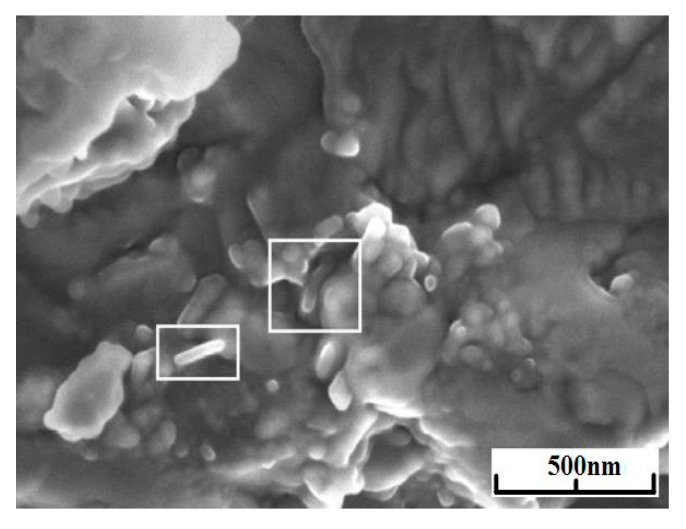
Micrograph of the fractures of samples of ceramics based on cubic BN doped with graphene flakes, synthesized in acetylene/argon plasma.

**Figure 16 materials-13-01728-f016:**
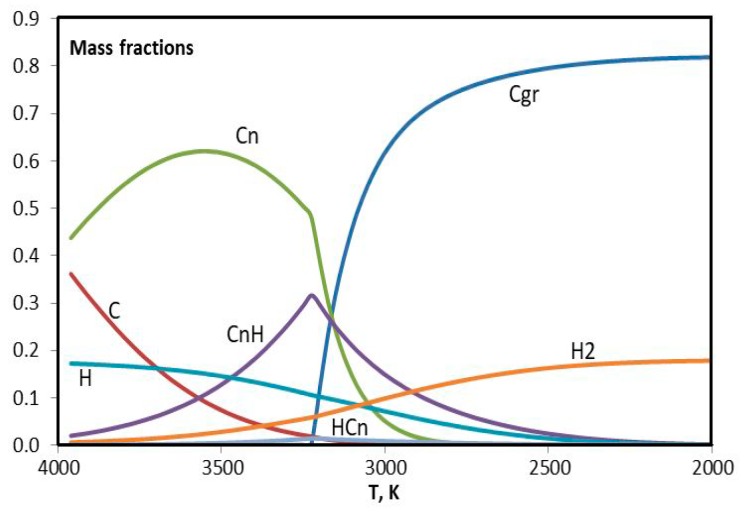
Evolution of the composition of helium plasma with addition of the propane-butane mixture in the temperature range 2000–4000 К.

**Figure 17 materials-13-01728-f017:**
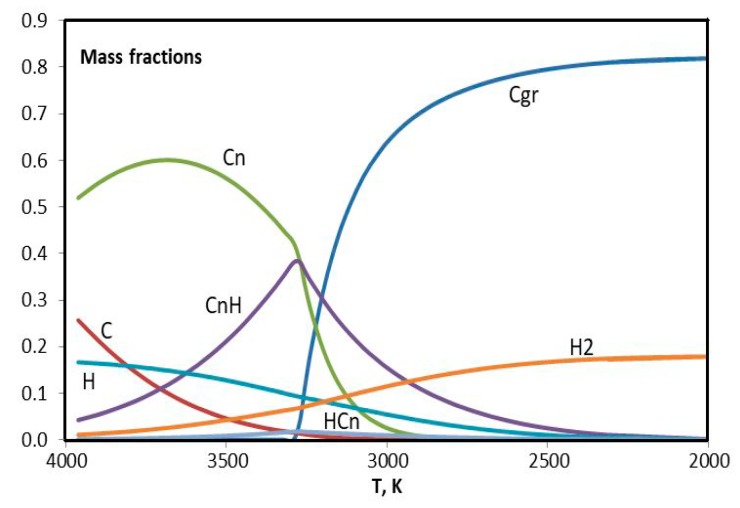
Evolution of the composition of argon plasma with addition of the propane-butane mixture in the temperature range 2000–4000 К.

**Figure 18 materials-13-01728-f018:**
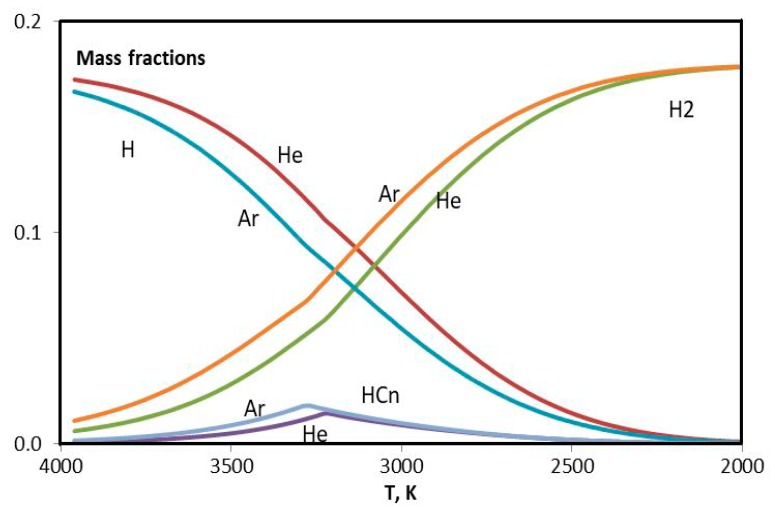
Evolution of H, H_2_, and HCn in the helium and argon plasma with addition of the propane-butane mixture in the temperature range 2000–4000 К.

**Figure 19 materials-13-01728-f019:**
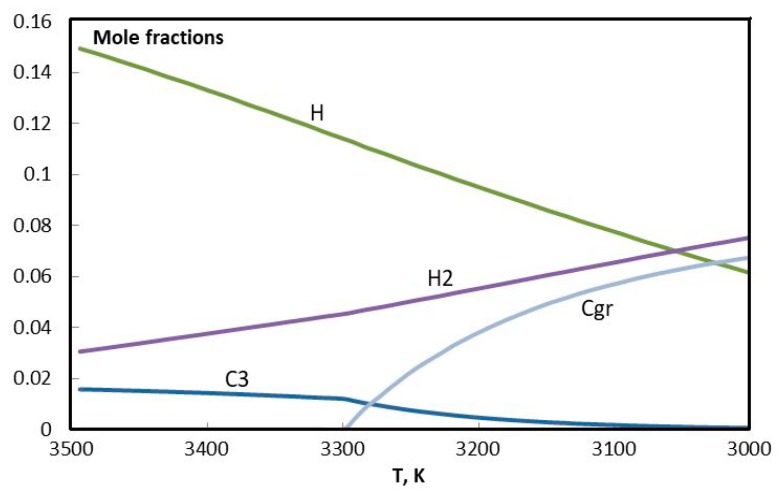
Evolution of the composition of argon plasma with addition of the propane-butane mixture in the temperature range 3000–3500 К (big fractions).

**Figure 20 materials-13-01728-f020:**
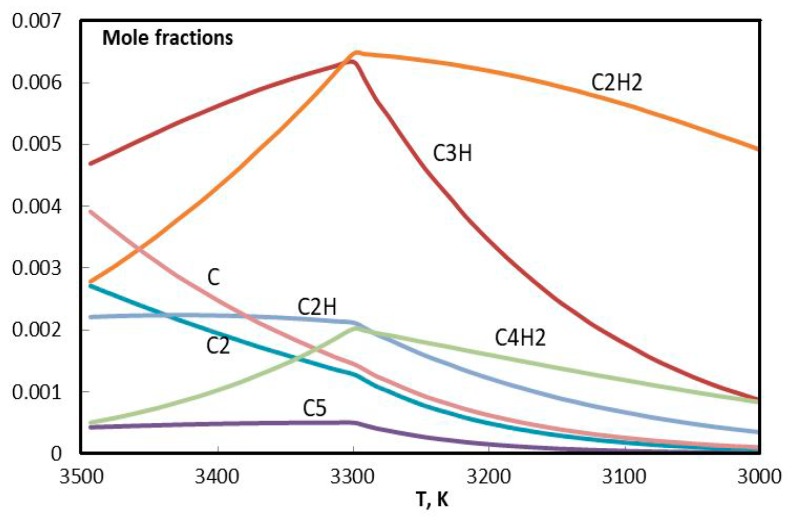
Evolution of the composition of argon plasma with addition of the propane-butane mixture in the temperature range 3000–3500 К (small fractions).

**Figure 21 materials-13-01728-f021:**
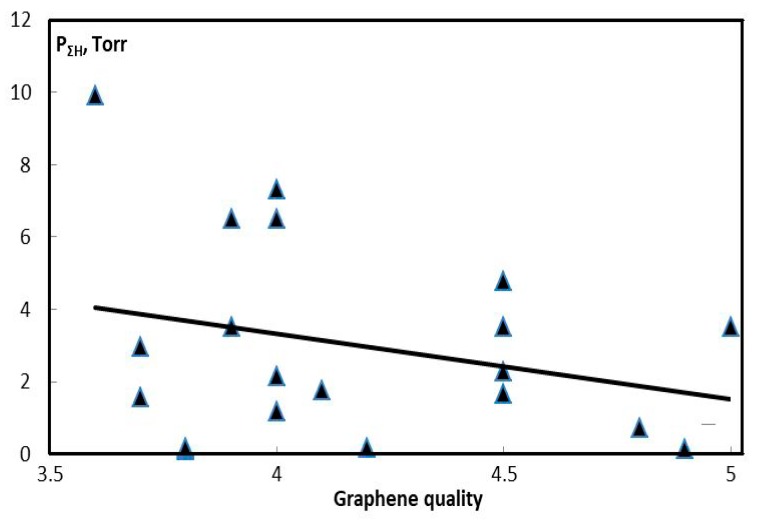
Statistical relation between the graphene quality and the sum of partial pressures of H and H_2_ at the beginning of condensation of carbon in the temperature history of the process.

**Table 1 materials-13-01728-t001:** Experimental conditions.

Power (kW)	Current (A)	Voltage (V)	Gas Pressure (Torr)	Helium Flow Rate (g s^−1^)	Argon Flow Rate (g s^−1^)	Propane-Butane Flow Rate (g s^−1^)	Methane Flow Rate (g s^−1^)	Acetylene Flow Rate (g s^−1^)
22–36	350–400	60–90	150–710	0.75–0.9	3.0–3.75	0.11–0.30	0.15–0.37	0.05–0.16

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
