# Peer review of "Distinctive Features of Graphene Synthesized in a Plasma Jet Created by a DC Plasma Torch"

_materials, 2020, doi:10.3390/ma13071728_

Round 1

Reviewer 1 Report

Check the language style and grammar eg.:

line 13 - precursorS
line 19 - hydrogen-rich
line 30 - quasi-infinitive
line 36 - graphene
line 40 - double-layer (and put "-" in other complex expressions. There are a lot of such expressions in the text)

Use indefinite and definite articles where necessary.
Put commas where necessary

Is there any time in the process as pre-treatment time (for example for reactor evacuation, for oxygen removement)?

Figure 2d is not visible and should be enlarged.
Figure 3 should be a little enlarged.

line 190 - Do you mean the atom or mass ratio?

Figure 4d is not visible and should be enlarged.
Figure 4 should be enlarged.
Figure 5 should be a little enlarged.

Figure 8 is not visible and should be enlarged.
Figure 9 should be enlarged.
Enlarge Figures 10b and 11b, and 12b

Figure 11 - in caption "argon" is in Cyryllic

Figure 9,10,11 - consider to use one scale to compare figures (Fig 9 is in nm Fig 11 in micrometers)

Figure 12 should be enlarged.

In figures describing thermogravimetric analysis there are essential data that are not visible. So numbers or all graphs should be enlarged.

line 242 - ceramics were
line 243 - in Figure
line 248 - related to
line 260 - is it corroect range? 4000-2000 K

Unify units - in some parts and Figures there are Celcius, in others there are Kelvins; in some parts thera are Torrs in some parts there are GPa)

unify chemical expressions. In some cases subscripts are used (e.g. H20 under Figure 10 and C2H@ in the line 265

What software or method was used to obtain modeling results in Figures 15,16, 17 and 18.
Enlarge Figures 15, 16, 17,18, 19, 20.

Some details about purity if the final material could improve the presented results.

line 305 - check grammar

Author Response

Much esteemed editorial staff!

Authors are thankful to reviewers for their intention to make manuscript better understandable. According to their notes we made changes in text marked with yellow.

To reviewer #1

  1. Check the language style and grammar eg.:

line 13 - precursorS

line 19 - hydrogen-rich

line 30 - quasi-infinitive, better “of quasi-infinite size”

line 36 - graphene

line 40 - double-layer (and put "-" in other complex expressions. There are a lot of such expressions in the text)

Use indefinite and definite articles where necessary.

Putcommaswherenecessary

We have tried to take into consideration all remarks.

  1. Is there any time in the process as pre-treatment time (for example for reactor evacuation, for oxygen removement)?

In the scope of pre-treatment the reactor usually was depressurized down to 77 Torr (bottom limit for the liquid-packed ring pump), the pressure subsequently being brought to the work level. The whole process takes several minutes.

  1. Figure 2d is not visible and should be enlarged. Is enlarged.
  2. Figure 3 should be a little enlarged.Is enlarged.
  3. line 190 - Do you mean the atom or mass ratio?

Text is changed:”obtained the hydrogen to carbon molar ratio of 1:4.”

  1. Figure 4d is not visible and should be enlarged. Figure 4 should be enlarged. Is enlarged.
  2. Figure 5 should be a little enlarged.Is enlarged.
  3. Figure 8 is not visible and should be enlarged. Figure 9 should be enlarged. Enlarge Figures 10b and 11b, and 12bAre enlarged.
  4. Figure 11 - in caption "argon" is in Cyryllic.
  5. Figure 9,10,11 - consider to use one scale to compare figures (Fig 9 is in nm Fig 11 in micrometers). Different magnifications are needed to make the morphology objectively comparable. Equal magnification is reasonable when changes of surface propertiesare in focus due to the impact or change of conditions of the synthesis leading to the change of morphology. In our case the graphene synthesized in plasma volume has a flake-like form independent of the jet composition. Different conditions have influence on their dimensions, on the existence of swelled domains (gases) or black insertions (anode metal).
  6. Figure 12 should be enlarged.Is enlarged.
  7. In figures describing thermogravimetric analysis there are essential data that are not visible. Sonumbersorallgraphsshouldbeenlarged. According to this remark all graphic results of thermal analysis are enlarged.
  8. line 242 - ceramics were line 243 - in Figure line 248 - related to line 260 - is it corroect range? 4000-2000 K. Temperature of the press operation was 1800K. The range 2000-4000K is a result of the modeling. In this range precursors of graphene structures are formed.

Text is included:

Ceramics were produced by means of the hot pressing ( 1800 К,412 MТоrr) and reinforced with carbon nanotubes, as one can see in Figure 14, at the fracture.

  1. Unify units - in some parts and Figures there are Celcius, in others there are Kelvins; in some parts thera are Torrs in some parts there are GPa)

Changesto Kelvins aremadeinthe figures and the text.

  1. unify chemical expressions. In some cases subscripts are used (e.g. H20 under Figure 10 and C2H@ in the line 265

Changed..

  1. What software or method was used to obtain modeling results in Figures 15,16, 17 and 18.

Method is discussed in detail in [28]:Shavelkina M.B.; Ivanov P.P.; Bocharov A.N.; Amirov R. Kh.  1D modeling of the equilibrium plasma flow in the scope of direct current plasma torch assisted graphene synthesis. J. Phys. D: Appl. Phys. 2019, 52, 49, 495202; DOI: 10.1088/1361-6463/ab407.

The same approach was used in:

[37]. Subramaniam V.; Underwood T.C.; Raja L.L.; Cappelli M.A. Computational and experimental investigation of plasma deflagration jets and detonation shocks in Coaxial plasma accelerators. Plasma Sources Science and Technology.2018, 27, 2. DOI:10.1088/1361-6595/aaabec.

[38].Kozlov A.N.; Konovalov V.S. Pulsating flow regimes of ionizing gas in coaxial plasma accelerators. KIAM Preprint M.V. Keldysh. 2014. 28 s. URL: http://library.keldysh.ru/preprint.asp?id=2014-1. [in Russian].

[39].Batenin V. M.; Bityurin V. A., Zhelnin V. A.; Ivanov P. P.; Medin S. A.; Lyubimov G. A.; Satanovsky V. R. ; Turovets V. L. Gas-dynamic and electrical characteristics of MHD generator according to data from physical and numerical experiments - RM channel of the U-25 device. HighTemp.1983, 21, 438-447; [inRussian].

  1. Enlarge Figures 15, 16, 17,18, 19, 20.Are enlarged.
  2. Some details about purity if the final material could improve the presented results.

Structures, synthesized in the volume of plasma, are formed actually by means of the upwards assemblage in an organized conditions. Process organization is realized by the variation of temperature profile. Process kinetics is fully controlled by the temperature. But what is uncontrollable, what doesn’t depend on the conditions of volumetric synthesis, that is particle morphology. In the real experiment without catalitic substrates it is impossible to get relatively smooth films of large dimensions. Graphene synthesis doesn’t occur due to the simple pulverization of carbon into vacuum. Graphene is synthesized in its peculiarities in the hydrogen-rich medium created by the DC plasma torch.

Inserted into the conclusion:

Raman spectroscopy shows a high content of defects in samples. Apparently hydrogen is responsible for the bent form of graphene. Hydrogen content in precursor and the type of carrier-gas are shown to influence on the lateral dimensions of graphene and its purity. Greatest output of pure graphene occurs when propane-butane mixture is added to the helium plasma jet, and maximal graphene dimensions are obtained using methane and argon. Thus it is possible to control graphene properties by means of the plasma forming gas. Variation of temperature profile in reactor is in direct correspondence with properties of plasma forming gas. At every time instance different plasma components are formed according to the temperature profile.

  1. line 305 –checkgrammar

Changed.

In the case of argon, the residence time of particles in the hydrogen enriched atmosphere is greater, so greater are

Reviewer 2 Report

The manuscript deals with the synthesis of graphene materials using a plasmas stream from a DC plasma torch. The resultant graphene derivative was characterised using Raman spectroscopy, mass-spectrometry and SEM to understand the structure and property relationship. The manuscript appears to be decently written but needs a revision to fix some basic flaws and errors.

  1. Raman data:
    1. IG/ID is wrong. It should ID/IG and should be written as ID/IG
    2. Whey dos the 2D peak vary in intensity over acetylene presence, and why is it sharper with methane? What is the chemistry and mechanism behind it?
    3. The spectra should be normalised and must be plotted in a stacked manner to compare different graphene derivatives.
    4.  

  1. Please take care of the chemical names - C3H should be C3 These kinds of errors should not be present.

  1. Thermal analytic graphs should be given with more clear way and quality. In general, all the figures should be supplied at maximum possible resolutions. Currently, there are of poor quality. SEM images should be improved in resolution.

  1. Did authors do a proper XPS analysis? It does not seem to be done properly. Where are the wide scan and individual peak spectra? Without which, the discussion will become invalid.

  1. Finally, what is the novelty of this work which is not well established in the introduction? There are plenty of works reported on a similar specific subject matter (synthesis of graphene using plasma). How does this work stand out?

  1. Formatting and other issues:

  • Occasional grammatical errors found: especially, the tenses change abruptly. Please take care of it.
  • Use the same font style in all the images.

Author Response

To reviewer #2

  1. Raman data:
  2. IG/ID is wrong. It should ID/IG and should be written as ID/IG.

This slip is corrected. Thanks.

  1. Whey dos the 2D peak vary in intensity over acetylene presence, and why is it sharper with methane? What is the chemistry and mechanism behind it?

This question might be answered by the solution of the integrated model including the two-dimensional chemical kinetics, and thermodynamics, and methods of molecular dynamics or Monte-Carlo. In the scope of this work we have tried to illustrate the experiment of graphene synthesis in the DC plasma jet by means of one-dimensional modeling, when graphene is formed in the hydrogen-rich medium under the influence of radicals and hydrogen resulting in “flakes” instead of film. So the difference in the intensity of D-peaks might be explained by the difference of concentrations of critical components on the graphene surface. In experiment it is the difference in results of thermal analysis.

  1. The spectra should be normalised and must be plotted in a stacked manner to compare different graphene derivatives.

We agree with reviewer, we tried to do it in the very beginning in order to analyse the data, but failed. There must be backtracking of one parameter. Goal of experiments was the study of morphology under variation of several factors: the pressure, precursors, carrier gases. Several graphs are needed. We left the subject for the future in anticipation of results in chemical kinetics. Something ispublishedalreadyin[M B Shavelkina, E A Filimonova, R KhAmirov Effect of helium/propane-butane atmosphere on the synthesis of graphene in plasma jet system // Plasma Sources Science and Technology, 2020,Volume 29, Number 2, DOI: 10.1088/1361-6595/ab61e].

2.Pleasetakecareofthechemicalnames - C3HshouldbeCThesekindsoferrors should not be present.

Is corrected.

  1. Thermal analytic graphs should be given with more clear way and quality. In general, all the figures should be supplied at maximum possible resolutions. Currently, there are of poor quality. SEM images should be improved in resolution.

Graphs of thermal analysis are enlarged, and symbols became better visible. Pictures are amended.

  1. Did authors do a proper XPS analysis? It does not seem to be done properly. Where are the wide scan and individual peak spectra? Without which, the discussion will become invalid.

We have XPS analysis spectra. Here below they are on one graph:

This method has the peculiarity – it “doesn’t see” hydrogen.The XPS method analyzes the binding energy of electrons emitted from the sample surface under the influence of CuKα, AlKαorMgKα. We can evaluate the share of sp3carbon only. In the near future we hope to get molecular spectra in addition to the spectra of C2, C, H, CN in order to analyze the process of plasma synthesis. The aim is to examine the plasma jet processes leading to the flake like form, in the hope to control them.

  1. Finally, what is the novelty of this work which is not well established in the introduction? There are plenty of works reported on a similar specific subject matter (synthesis of graphene using plasma). How does this work stand out?

It is known that plasma jets generated by high power DC plasma torches provide high productivity of plasma chemical reactor. Only three groups of researchers from US, Canada  (Mendoza-Gonzalez NY, KimK.S.)and France are engaged in the synthesis of carbon nanostructures, mostly nanotubes, in the arc discharge. C’esttout. There is no references dealing with the synthesis of graphene and nanotubes in the jet from DC plasma torch. Microwave plasma, RF-plasma and the jet from DC plasma torch have substantially different process mechanisms. Besides, never before the topic was discussed why the free synthesis makes the flake like graphene, and why its morphology, depending on the reaction conditions, may vary from twisted to nearly strait sheet with bent edges. The difference in morphology makes different thermostability and different Raman spectra but the same three modes. And why hydrogen or its greater or lesser content is responsible for the graphene form. In the last analysis this form is beneficial for the physical-mechanical properties of ceramics when graphene is used as addition. We investigated the influence of flakes on the electrical characteristics of supercondensers, fuel cells, investigated their sorption properties. It is impossible to cover all topics in one paper, the main goal of which is to show the peculiarities of graphene synthesized in the jet from DC plasma torch.

  1. Formattingandotherissues:

Occasional grammatical errors found: especially, the tenses change abruptly. Please take care of it.

Thanks. We have tried to correct all.

Usethesamefontstyleinalltheimages.

We have tried to do it. But in this case the information value seems to be lost. Mainly we tried to bring to reader results of different methods of investigation.

Round 2

Reviewer 2 Report

The manuscript has been revised to an extent. However, not to the entire  satisfaction based the comments provided earlier:

  1. Author included the widescan spectra, C1s and O1s in the response document, but not in the actual manuscript.
  2. They should also process the data properly with appropriate curve fitting. Then include them in the manuscript.

Until then, I would hesitate to accept this manuscript for publication.

Author Response

According to the notes of reviewer #2 we made changes in text marked with yellow. Full answer follows in DOC-file attached
